



# Statistical assessment of a Doppler radar model of TKE dissipation rate for low Richardson numbers (weakly stratified or strongly sheared conditions)

Hubert Luce[1], Lakshmi Kantha[2], Hiroyuki Hashiguchi[1]

[1]Research Institute for Sustainable Humanosphere, Kyoto University, Kyoto, 611-0011, Japan
[2]Smead Aerospace Engineering Sciences, University of Colorado, Boulder, CO, USA

*Correspondence to*: Hubert Luce (luce@rish.kyoto-u.ac.jp)

**Abstract.** The potential ability of VHF or UHF Doppler radars to measure Turbulence Kinetic Energy (TKE) dissipation rate ε

in the atmosphere is a major asset of these instruments, because of the possibility of continuous monitoring of turbulence in

the atmospheric column above the radars. Several models have been proposed over past decades to relate ε to the half width $\sigma$

of the Doppler spectrum peak, corrected for non-turbulent contributions, but their relevance remains unclear. Recently, Luce

et al. (2023) tested the performance of a new model expected to be valid for weakly stratified or strongly sheared conditions,

i.e. for low Richardson (Ri) numbers. Its simplest expression is $\varepsilon_S = C_S \sigma^2 S$ where $C_S \sim 0.64$ and $S = \left| d\vec{V}/dz \right|$ is the vertical

shear of the horizontal wind $\vec{V}$. We assessed the relevance of this model with a UHF (1.357 GHz) wind profiler called WPR

LQ-7, which is routinely operated at Shigaraki Middle and Upper Atmosphere (MU) observatory (34.85°N, 136.10°E) in

Japan. For this purpose, we selected turbulence events associated with Kelvin-Helmholtz (KH) billows, because their formation

necessarily requires $Ri < 0.25$ somewhere in the flow, a condition *a priori* favorable to the application of the model. Eleven

years of WPR LQ-7 data were used for this objective. The assessment of $\varepsilon_S$ was first based on its consistency with an empirical

model $\varepsilon_{Lout} = \sigma^3/L_{out}$ that was found to compare well in a KH layer with direct estimates of $\varepsilon$ from in-situ measurements

for $L_{out} \approx 70 \, m$. Some degree of equivalence between $\varepsilon_S$ and $\varepsilon_{Lout}$ was confirmed by statistical analysis of 192 KH layers

found in the height range [0.3-5.0] km indicating that $L_{out} \approx L_H/0.64$ where $L_H = \sigma/S$ is the Hunt scale defined for neutral

turbulence. The degree of equivalence is even significantly improved if $L_{out}$ is not treated as a constant but depends on the

depth $D$ of the layer. We found $L_{out} \approx 0.0875 \, D$ or equivalently $L_H \sim 0.056 \, D$ which also means that $\sigma$ is proportional to the

apparent variation of the horizontal velocity ($S \times D$) over the depth of the KH layer. Consequently, $\varepsilon_S = 0.64 \, \sigma^2 S$ and $\varepsilon_{Lout} =$

$\sigma^3/0.0875D$ would express the same model for KH layers when $Ri$ remains small. For such a condition, we provide a physical

interpretation of $\varepsilon_{Lout}$, which would be qualitatively identical to that for neutral boundary layers.



## 1 Introduction

VHF Stratosphere-Troposphere (ST) radars and UHF wind profilers can be used to estimate turbulence kinetic energy (TKE)
dissipation rate $\varepsilon$ in the atmosphere, from half the Doppler spectral width corrected for non-turbulent effects (hereafter, denoted
by $\sigma$) (e.g. *Hocking, 1983; Doviak and Zrnic', 1984; Hocking et al., 2016*, and references therein). If the measurements are
made with a zenith-pointing beam as is the case in the present paper, $\sigma^2$ is expected to be an estimate of the variance $\langle w'^2 \rangle$ of
the vertical component of wind fluctuations produced by turbulence. Luce et al. (2023) (hereafter L2023) tested three radar
models relating $\sigma^2$ to $\varepsilon$ using data collected by a UHF wind profiler called WPR-LQ-7 (Imai et al., 2007), and routinely
operated at Shigaraki MU Observatory (34.85°N, 136.10°E) in Japan. The models require determination of the non-
dimensional gradient Richardson number $Ri = N^2/S^2$, where $S = \left| d\vec{V}/dz \right| (s^{-1})$ is the vertical shear of the horizontal wind,
and $N (rads^{-1})$ defined by $N^2 = (g/\theta) \, d\theta/dz$, where $\theta$ is the potential temperature and $g$ is gravitational acceleration, is
the Brünt-Vaïsälä or buoyancy frequency. Comparisons with direct estimates of $\varepsilon$ obtained from in-situ measurements made
using turbulence sensors aboard fixed-wing Unmanned Aerial Vehicles (UAVs called DataHawks, *Lawrence and Balsley,
2013; Kantha et al., 2017*) in the altitude range 0.3-4.5 km, revealed two findings. (1) An empirical model $\varepsilon_{Lout} = \sigma^3/L_{out}$
with $L_{out} \sim 70$ m provides the best statistical agreement with *in-situ* estimates (Figure 5 of L2023) confirming results obtained
by Luce et al. (2018) (their Figure 9) with the VHF MU radar. (2) The model $\varepsilon_S = C_S\sigma^2 S$ proposed originally by *Hunt et al.
(1988)* from heuristic arguments and confirmed by *Basu et al. (2021)* and *Basu and Holtslag (2022)* from Direct Numerical
Simulations (DNS) and analytical derivations (and expected to be valid for weakly stratified and/or strongly sheared
conditions, i.e. for low $Ri$ values), provides better agreement than $\varepsilon_N = C_N\sigma^2 N$, commonly and extensively used and expected
to be applicable to turbulence under stable stratification *(e.g. Hocking, 1983; Hocking et al., 2016)*. Unlike $\varepsilon_S$, no conditions
on $N$ or $Ri$ have been established for $\varepsilon_N$ to be valid (except that $N^2$ must be positive). Clearly, $\varepsilon_N$ is not valid for neutral
stratification.

L2023 showed that $\varepsilon_S$ gives statistical results that are "intermediate" to $\varepsilon_{Lout}$ and $\varepsilon_N$. As turbulence is expected to occur and
to be maintained mainly when $Ri$ is low, this property should favor the validity of $\varepsilon_S$ over $\varepsilon_N$, if $\varepsilon_S$ is a relevant model. To
ascertain this, we need to check the model under the conditions for which it is supposed to be valid, i.e. when $Ri$ is low (less
than roughly 0.2), according to DNS of *Basu et al. (2021)*. $Ri$ cannot be estimated from the radar data alone, because $N^2$ is
not measurable by radar. It is usually obtained from measurements of pressure and temperature and winds by meteorological
radiosondes. However, radiosonde measurements are scarce and rarely co-located with radar measurements. In addition, $Ri$ is
a scale-dependent parameter (e.g. *Balsley et al., 2008*) and there is no prescribed method to calculate the appropriate value of
$Ri$, making it difficult to apply quantitative criteria on $Ri$ for a selection of cases to be studied. For the present study, we used
an alternative strategy that avoids these difficulties, the goal being to find out if, not when, $\varepsilon_S$ can actually be relevant for low
$Ri$ values. The most favorable condition for this goal is Kelvin-Helmholtz (KH) billows. Indeed, these structures are produced
by shear instabilities, which grow when $Ri < Ri_c = 0.25$ is met somewhere in the flow and are generally associated with
enhanced turbulence. They are also clearly visible in radar echoes, and hence easily identified and earmarked for further study.





L2023 evaluated the performance of the radar models for a KH layer (i.e. a turbulent layer exhibiting KH billows) of ~800 m in depth sampled several times by a DataHawk during the 2017 Shigaraki UAV Radar Experiment (ShUREX). They found that both $\varepsilon_{Lout}$ with $L_{out} \sim 70\ m$ and $\varepsilon_S$ provide values consistent with DataHawk-derived $\varepsilon$. This result is the cornerstone of the reasoning that we will follow in this paper. If it is representative, this would provide a physical interpretation of the empirical model when applied to turbulence produced by a KH instability. Therefore, we tried to answer the following question: To what extent is the equivalence between $\varepsilon_S$ and $\varepsilon_{Lout}$ also observed for other KH events? For this purpose, we searched for KH layers in time-height cross-sections of WPR LQ-7 echo power from 2011 to 2021. We identified 192 cases that could be easily analyzed. They allowed us to verify and qualify the result stated by L2023 by taking into account the influence of the depth $D$ of the KH layers and to infer a relationship between the Hunt scale $L_H$ defined as $\sqrt{\langle w'^2 \rangle}/S$ and $D$.

Section 2 describes the main characteristics of WPR LQ-7 and the parameters used for routine observations. Section 3 briefly introduces the $\varepsilon_S$ model and the results of the case study described by L2023. Section 4 presents the method and criteria used for the KH layer selection. Section 5 shows the statistical results for 192 KH layers identified, and for 113 turbulent KH layers selected more subjectively for analysis. Finally, conclusions are given in Section 6.

## 2. The WPR-LQ-7

The WPR LQ-7 is a 1.3575 GHz Doppler radar. It has a phased array antenna composed of seven Luneberg lenses of 800 mm diameter (Figure 1). Its peak output power is 2.8 kW. It can be steered into five directions sequentially (i.e. after FFT operations), vertical and 14.2° off zenith toward North, East, South and West. The main radar parameters of the WPR-LQ-7, installed and operated at Shigaraki MU Observatory since 2006, are given in Table 1. The acquisition time for one profile composed of 80 altitudes from 300 m AGL every 100 m in each direction after 18 incoherent integrations is 59 sec, but for a total of 11.8 sec of observations for each direction (due to the intertwining between the directions). The time series are processed by automatic algorithms to remove outliers (e.g., bats, birds, airplanes) and ground clutter as far as possible. Low signals near and below the detection thresholds are removed, and profiles of atmospheric parameters (echo power, radial winds, half-power spectral width, horizontal and vertical winds) averaged over 10 min are made available for routine monitoring (http://www.rish.kyoto-u.ac.jp/radar-group/blr/shigaraki/data/). Because of the high data quality control, the 10-min averaged data are used to retrieve $\varepsilon$ with the goal of assessing these routine data for further analyses.





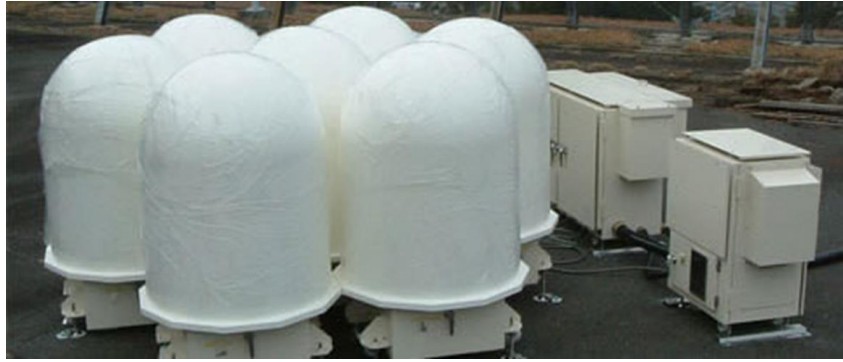


**Figure 1**. **WPR LQ-7 Luneberg Lens antenna array.**

| Parameter | |
|---|---|
| Beam directions | (0°,0°),(0°,14.2°), (90°,14.2°), (180°,14.2°), (270°,14.2°) |
| Radar frequency (MHz) | 1357 |
| Interpulse period ($\mu s$) | 80 |
| Subpulse duration ($\mu s$) | 0.67 |
| Pulse coding | 16-bit optimal complementary code |
| Range resolution (m) | 100 |
| Number of gates | 80 |
| Coherent integration number | 64 |
| Incoherent integration number | 18 |
| Number of FFT points | 128 |
| Acquisition time for one profile (s) (Antenna beam switched after FFT) | 59 s |
| Acquisition time of the mean profile (min) | 10 |
| Velocity aliasing ($ms^{-1}$) | 10.8 |

**Table 1: WPR-LQ-7 parameters in routine observation mode**


## 3. The $\varepsilon_S$ model and its application to a case study

### 3.1 Description

For low gradient Richardson numbers, several studies showed from heuristic arguments that $\varepsilon$ can be written as (*Hunt et al., 1988; Schumann and Gerz, 1995*):



$$\varepsilon_S = C_S \langle w'^2 \rangle S \tag{1}$$

where $\langle w'^2 \rangle$ is the variance of the vertical wind fluctuations produced by turbulence and $C_S$ is a constant. Expression (1) is equivalent to $\varepsilon_S = C_S \langle w'^2 \rangle^{3/2}/L_H$ where $L_H = \sqrt{\langle w'^2 \rangle}/S$ is the Hunt scale. The Hunt scale describes the maximum size of the turbulent eddies, which are stretched and destroyed by the wind shear. In strongly sheared or weakly stratified flows, eddies can be affected first by the wind shear before being affected by the stratification. *Hunt et al. (1988)* suggested that

Expression (1) can be valid up to $Ri \sim 0.5$. *Schumann and Gerz* (1995) found that it can be valid up to $Ri \sim 1$ from LES. From simplified budget equations for TKE and temperature variance, and using similarity theory, *Basu and Holtslag* (2022) evaluated the constant $C_S$ to be 0.64 and provided a generalization of (1):

$$\varepsilon' = 0.64\left(1 - R_f\right)^{1/2} \langle w'^2 \rangle S \tag{2}$$

where $R_f$ is the flux Richardson number. For $Ri \to 0$, $R_f \to 0$, then $\varepsilon' \to \varepsilon_S = 0.64\langle w'^2 \rangle S$, i.e. Expression (1) with $C_S = $

0.64. From DNS, *Basu et al.* (2021) found expression (1) with $C_S = 0.60$ for $Ri$ up to 0.2 at least. L2023 put in perspective $\varepsilon_S$ and the commonly used model $\varepsilon_N = 0.5\langle w'^2 \rangle N$ which can be re-written as $\varepsilon_N = 0.5\langle w'^2 \rangle^{3/2}/L_B$ where $L_B = \sqrt{\langle w'^2 \rangle}/N$ is the buoyancy scale. $L_B$ is a measure of the eddy scale at which vertical turbulent motions are suppressed. By definition, when $Ri < 1$, $L_H < L_B$ and vice-versa. It is then quite logical to assume that, when $Ri \ll 1$, stratification effects can be neglected and $\varepsilon_S$ can be more appropriate than $\varepsilon_N$. In Appendix, we propose the corresponding expression of heat diffusivity

for small $Ri$ values, when Expression (2) is valid.

If $\sigma^2$ can be associated with $\langle w'^2 \rangle$, then $\varepsilon_S$ can be evaluated from the radar data alone, since an estimate of $S$ can be obtained at the range and time resolutions of the radar. Expression (1) was applied by *Fukao et al.* (2011) to KH layers detected by the 46.5 MHz MU radar, but with a coefficient different from 0.64 and not for the right reason. This was to compensate for the lack of $N^2$ measurements, assuming *that* $\varepsilon_N$ was the appropriate model.

## 120 3.2 The case study

Figure 2a shows the time-height cross-section of Signal to Noise Ratio SNR (dB) at vertical incidence on June 18, 2017 from 13:30 LT to 16:30 LT and between 0.3 km and 7 km altitudes (ASL) at a time resolution of ~1 min. A KH layer of about 800 m in depth associated with enhanced SNR is clearly visible between of 3.0 and 4.0 km altitudes, and between 15:00 LT and 16:00 LT. This case was analyzed in detail by L2023. The layer was crossed four times by a DataHawk, whose distance versus

time is highlighted in red in Fig. 2a. Figure 3 is the same as Figure 2 of L2023 but restricted to information pertinent to the objective of the present work. It is shown here again because it is the cornerstone of this paper and makes it self-sufficient. The DataHawk data processing applied to retrieve profiles of $\varepsilon$ can be found in *Luce et al.* (2018). The 20-min averaged profiles of $\varepsilon_S$ (solid red) and $\varepsilon_{Lout}$ (solid green) with $L_{out} = 70\,m$ (left panel of Figure 3) are nearly identical and coincide well with the four DataHawk-derived $\varepsilon$ profiles in the altitude range of the KH layer. The $Ri$ profile calculated at a vertical





resolution of 20 m from data collected by a Vaisala RS92-SGP radiosonde launched shortly after the DataHawk (right panel of Figure 3) presents a minimum consistent with a shear flow instability in the altitude range of the identified KH layer. However, it is also variable and shows negative minima. The mean value of $Ri$ over the depth of the KH layer is 0.09 and thus less than 0.2. But nothing tells us that this value is the one we should really consider to test the validity of the model. In addition, L2023 showed that the mean value of $Ri$ is 0.33 if calculated at a vertical resolution of 100 m and that the radiosonde

likely passed through the KH layer in a region where it was thinner, like the DataHawk during its first ascent. Therefore, problems related to the establishment of quantitative and objective criteria on the representativeness of the balloon measurements, on the estimation method of $Ri$, on the vertical resolution to be applied and on the selection of thresholds will be sources of major uncertainties, which can affect the statistical results. A selection based solely on the detected KH billows may be more reliable, even though there is no guarantee that $Ri$ remains below 0.25 when detected. The proposed approach is

validated *a posteriori*.

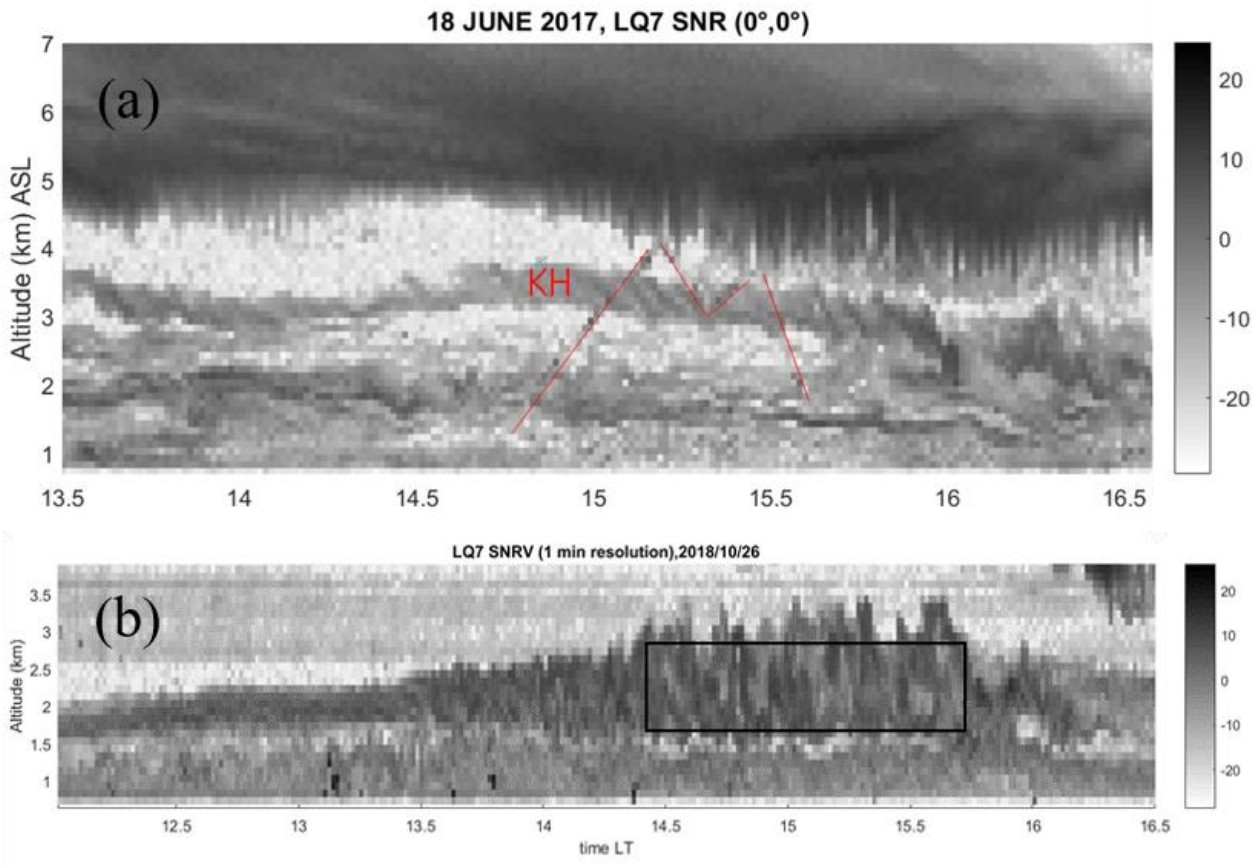

**Figure 2. Time-height cross-section of WPR LQ-7 SNR (dB) at vertical incidence (a) on 18 June 2017 from 13:30 LT to ~16:30 LT (b) on 26 October 2018 from 12:00 LT to 16:30 LT. Lines in (a) show the track of UAV DataHawk. The black rectangle in (b) shows**

**the time-altitude domain selected for analysis.**





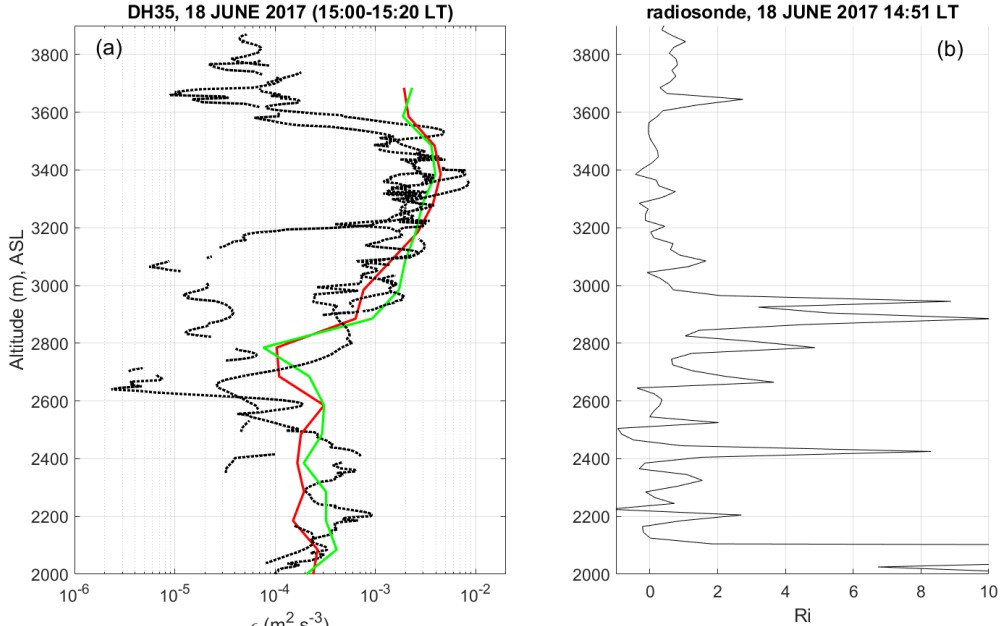

Figure 3. (a) DataHawk-derived $\varepsilon$ $(m^2 s^{-3})$ profiles in the height range 2,000 to 3,900 m during the ascents and descents of DataHawk (DH35) on 18 June 2017 (dotted black); $\varepsilon_S$ profile (solid red), $\varepsilon_{Lout}$ profile (solid green) derived from WPR LQ-7 radar data between 15:00 and 15:20 LT. (b) Richardson number profile estimated at a vertical resolution of 20 m from a simultaneous radiosonde (called V6). (From L2023)

The equivalence between $\varepsilon_S$ and $\varepsilon_{Lout}$ (Figure 3). i.e.:

$$0.64\, S \approx \sigma/L_{out} \tag{3}$$

was obtained with $L_{out} \approx 70\ m$ which had been found to be the canonical value of $L_{out}$ from the statistical comparison with 90 DataHawk-derived $\varepsilon$ profiles (Figure 4 of L2023). This is likely a coincidence, since $L_{out}$ cannot be treated as a constant. However, if $L_{out}$ is imposed to be constant, then we get $\sigma \sim S$, which is consistent with the fact that only the shear acts to generate TKE in a neutrally stratified flow. For neutral turbulence, in particular in boundary layers, turbulence length scales are proportional to the depth of the layer (e.g. Zilitinkevich et al., 2019). Therefore, it is expected that $L_{out}$ is related to the KH layer depth for weak stratification.

## 4. Method and criteria used for the KH layer selection



KH layers were first visually identified from 1-min resolution height-time cross-sections of SNR (dB) from 0.3 km to 5.0 km.

Eleven years of data (2011-2021) were screened in 12-hour segments. Structures similar to those observed in Fig. 2a were selected, sometimes with the help of the corresponding vertical velocities for confirmation, since KH billows are generally associated with vertical wind disturbances of periods/wavelengths identical to KH billows (e.g. *Klostermeyer and Rüster, 1981, Fukao et al., 2011* and references therein). Importantly, the selection criteria do not include wind shear and Doppler spectral width, since they are part of the parameters to be analyzed.

Turbulent layers rejected were:

(a) All cases that could be confused with convective instabilities at the top of the planetary boundary layer and at the edge of clouds or in precipitating clouds (generally associated with "smooth" echoes),

(b) All periods during which rain echoes were observed,

(c) Complex structures showing splitting or merging of echo layers or sporadic appearance (extremely frequent),

(d) Layers for which the depth was difficult to identify due to adjacent layers of enhanced echoes,

(e) Layers or part of layers showing a rapid change in depth or in altitude (because difficult to select with the method described below).

The 10-min averaged values of spectral width and winds were selected using a Matlab program allowing a manual selection of the layer with the mouse in a rectangle of dimensions representative of the duration and depth of the layer, when altitude

and depth were nearly constant and echo power did not change significantly. The depth of the KH layer was defined as the average of the maximum and minimum depths of the KH braids, also selected manually. The same KH event persisting for long but slowly moving in altitude and showing temporary fading may have been selected several times.

Our analysis cannot be considered as a statistical study of the occurrence of KH instabilities in the lower troposphere, because many of them may have been overlooked unintentionally, due to the absence of clearly visible KH braids at the stage of

evolution of the layer or due to insufficient time and/or range resolution. In particular, their occurrence seems to decrease quickly with height (not shown), because SNR decreases (blurring effect) and because the wind speed increases (under-sampling effect).

Figure 2b shows one of the deepest KH layers (~1500 m in average between 14:30 LT and 15:45 LT) among those selected during the eleven years of data. The portion selected for the analysis is shown by the black rectangle. This event is not

representative and shows unusual complex structures that may result from the successive development of several KH instabilities of different scales.

## 5. Statistical analyses

## 5.1 Analyses of the KH layers





Figure 4 shows the histogram of the depth D of the selected KH layers. The mean value is ~600 m and $D$ exceeds 300 m for

96% of them. Thinner KH layers are difficult or even impossible to identify due to range resolution limitation (100 m). The

KH layer described in Section 3.2 (~800 m in depth) is in the upper part of the distribution. 80% of the cases have a selected

duration between 30 min and 270 min (which is not the total duration of the event).

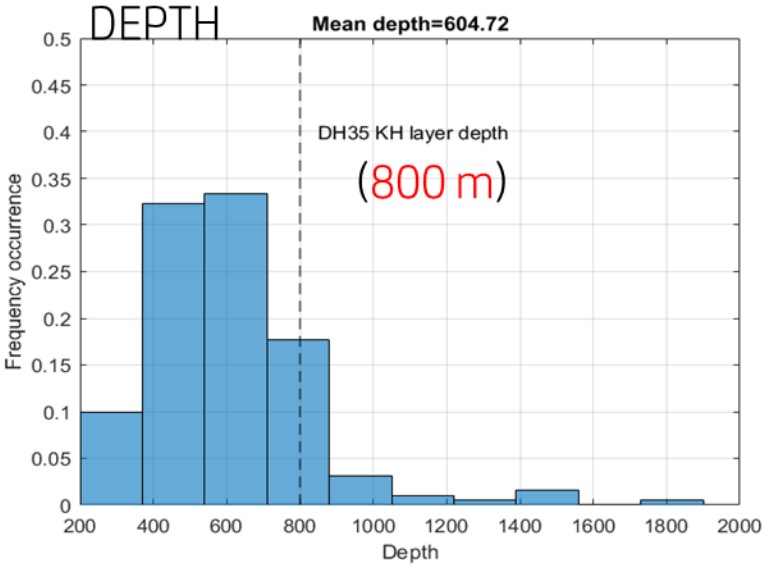

**Figure 4. Histogram of the depth of the 192 selected KH layers. The depth of the KH layer detected by DH35 (800 m) shown in**

**Figure 2a is indicated by the vertical dashed line.**

Figure 5 shows the scatter plot of $log_{10}(0.64\,S)$ vs $log_{10}(\sigma/L_{out})$ using all data for each case. For example, a layer selected

between 10:00LT and 10:20 LT and between 1000 m and 1500 m will contribute to a maximum of $(3 \times 6) = 18$ values, if

$\sigma$ and $S$ are defined everywhere in the rectangle. Figure 6a shows the same information, after taking the median (or without

substantial differences, the mean) value of all the estimates of $\sigma$ and $S$ in the time and height of the selected rectangles.

The best agreement in mean level between the two parameters was obtained for $\langle L_{out} \rangle = 50$ m in both cases, i.e. slightly less

than the canonical value (70 m). In Fig. 5, the dispersion of the distribution is large, but 74% of the disagreements are less than

a factor of 2 for a dynamic of values over a decade. The correlation coefficient is 0.25 only but significant according to the P

value (=0) and the regression slope is ~1. In Fig. 6a, the dispersion is less important (87% of the disagreements are within a

factor of 2). The correlation coefficient increases to 0.41, while the regression slope decreases (0.88). Therefore, Fig. 6a reveals

a basic trend between $\sigma$ and $S$, less obvious at shorter time and range resolutions, likely because of multiple sources of

uncertainties rather than due to a flaw in the assumption.

The influence of the layer depth can be shown in the following way: The ratio $< L_{out} >/70$ is close to $< D >/800$ suggesting

that $L_{out}$ is proportional to the depth of the KH layer, i.e., $L_{out} \approx 0.0875D$. Fig. 6b shows the scatter plot of $log_{10}(0.64\,S)$

vs $log_{10}(\sigma/0.0875D)$. The correlation coefficient significantly increases to 0.67 and the regression slope is 0.94. 96% of the





disagreements are now within a factor of 2. This is very consistent with the fact that $L_{out}$ should depend on the depth of the layer for low $Ri$ values. Figure 7a (7b) shows the concatenated values of $log_{10}(0.64\,S)$ (red line) and $log_{10}(\sigma/50)$ ($log_{10}(\sigma/0.0875D)$) (black line). The interdependence between the three parameters $(\sigma, S, D)$ is clear, in particular for the KH events 110 to 140. Then, we get:

$$L_H \approx 0.056\,D = D/17.9 \tag{4}$$

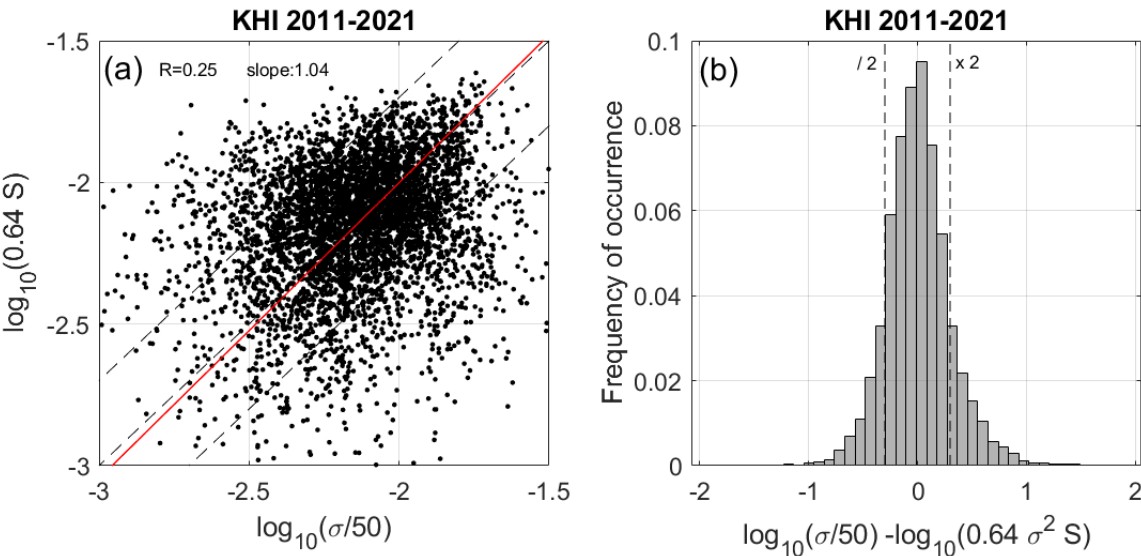

**Figure 5. (a) Scatter plot of $log_{10}(0.64\,S)$ vs $log_{10}(\sigma/L_{out})$ with $L_{out} = 50\,m$ for all KH layers at time and height resolutions of 10 min and 100 m, respectively. $R$ is the correlation coefficient. The regression line is shown in red. The slope is given in the insert. (b) Histogram of the difference between $log_{10}(0.64\,S)$ and $log_{10}(\sigma/L_{out})$.**




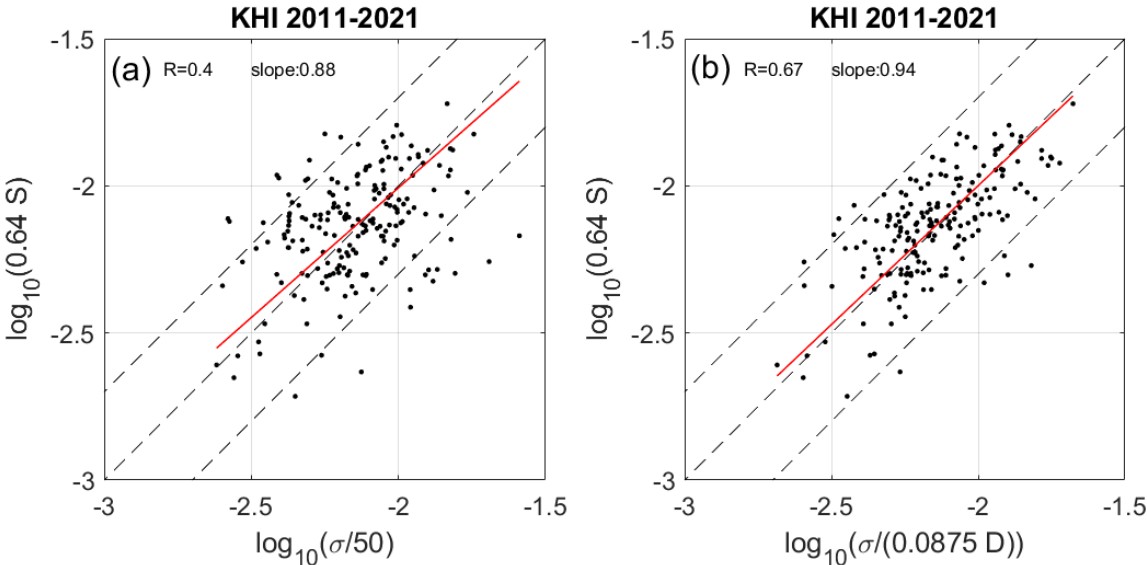

**Figure 6. (a) Same as in Fig. 5 after averaging all the values of $\sigma$ and $S$ in time and height domain of the selected rectangles. (b) Same as (a) with $L_{out} = 0.0875\ D$.**

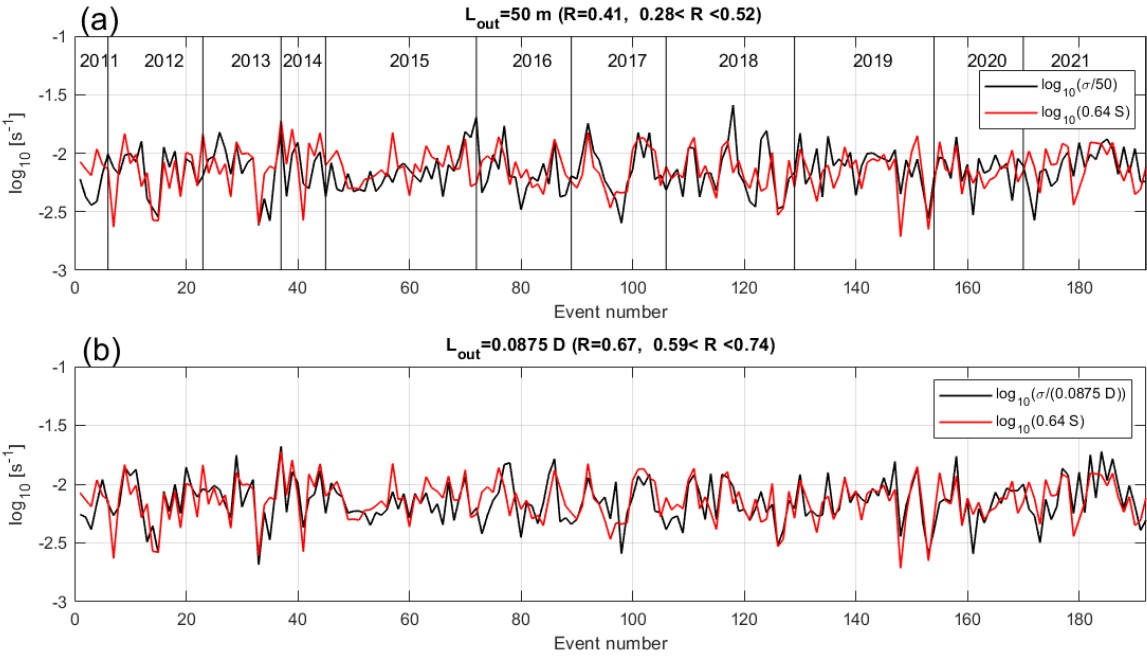

**Figure 7. (a) Time series of median values of $log_{10}(0.64\ S)$ (red line) and $log_{10}(\sigma/50)$ (black line) for the 192 KH layers. (b) Same as (a) for $L_{out} = 0.0875\ D$. $R$ is the correlation coefficient with its lower and upper bounds for a 95% confidence interval.**




In Table 4 of L2023, $L_H = \sigma/S$ for the KH layer was found to be 42 m for $D = 800$ m, fully consistent with the above

expression, since $\varepsilon_S$ and $\varepsilon_{Lout}$ were found to be equivalent. Figure 8 shows the linear relationship between $L_H$ or equivalently

$L_{out}$ and $D$ and an estimate of the slope from a linear regression for all data and for $D < 1000\, m$, because very few layers

have $D > 1000\, m$.

Figure 9 shows the scatter plot of $log_{10}(\varepsilon_S)$ vs $log_{10}(\varepsilon_{Lout})$ for $L_{out} = 50\, m$ (left panel) and $L_{out} \approx 0.0875D$ (right panel).

Due to the multiplication by $\sigma^2$, a strong self-correlation is introduced. The purpose of the figure is to show that, in practice,

the equivalence of the two models for KH layers would likely not produce different statistical results if compared with *in-situ*

(DataHawk) measurements.


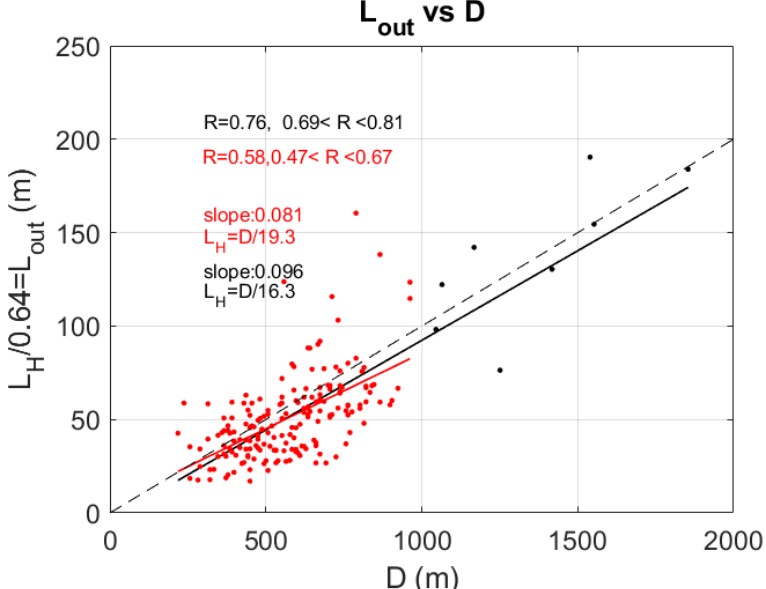

**Figure 8. Scatter plot of $L_{out} = L_H/0.64$ vs $D$. Regression lines and slopes are given for all the data (black) and for D<1000 m (red).**



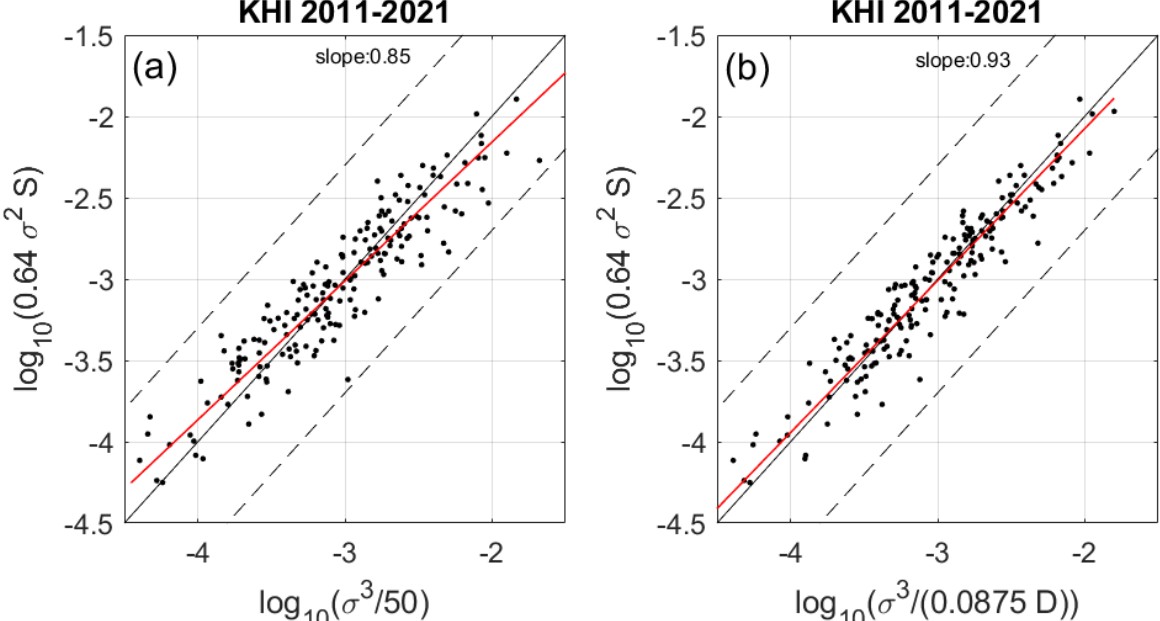

**Figure 9. (a)** Scatter plot of $log_{10}(\varepsilon_S)$ vs $log_{10}(\varepsilon_{Lout})$ with $L_{out} = 50\ m$. **(b)** Scatter plot of $log_{10}(\varepsilon_S)$ vs $log_{10}(\varepsilon_{Lout})$ with $L_{out} = 0.0875\ D$.

## 5.2 Application to other layers

Figures 10 to 12 show the same information as Figs 4, 6 and 7 for arbitrarily selected layers using the same method and rejection criteria as described in Section 5.1. The only difference is that they do not show evidence of KH braids, either because the layers were observed after the total collapse of the KH billows, or because they were generated by a different process, or because the KH billows were totally blurred by the insufficient time and range resolutions. The objective is to determine to what extent the results obtained for KH layers are also valid for unspecified layers. An arbitrary number of 113 layers was selected from 2017 data.

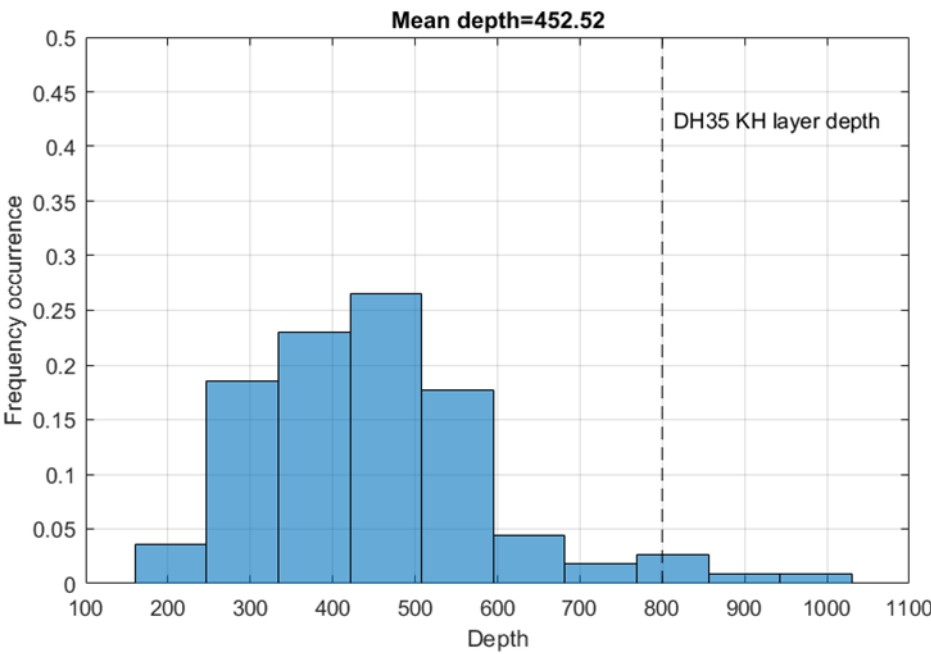

**Figure 10. Histogram of the depth of the 52 arbitrarily selected layers. The depth of the KH layer detected by DH35 (800 m) shown in Figure 2a is indicated by the vertical dashed line.**

The selected layers are in average thinner: ~450 m (Fig. 10). The best agreement in average between $log_{10}(0.64\,S)$ and $log_{10}(\sigma/L_{out})$ was obtained for $L_{out} \approx 35\,m$ (Fig. 11a), in full accordance with $L_{out} \approx 0.0875D$. The decrease of the dispersion with $L_{out} \approx 0.0875D$ (Fig. 11b) is not as important as in Fig 7b and the regression slope does not produce a satisfying trend. The correlation coefficient is lower but increases from 0.4 to 0.53 (Fig. 12). Part of the degradation of the results with respect to KH layers can be due to the increase of the difficulty to define the layer depth with accuracy, especially

for the thinnest ones. But it can also be due to the fact that the hypothesis of equivalence between $\varepsilon_S$ and $\varepsilon_{Lout}$ can sometimes be faulty if $Ri$ is not low. In L2023, the equivalence was not verified for a layer with $Ri \sim 1$.

Because the results are mixed, their quality can be considered as fair, if the objective is to get rough estimates of dissipation rates for climatological studies, and likely insufficient if precise estimates are necessary.





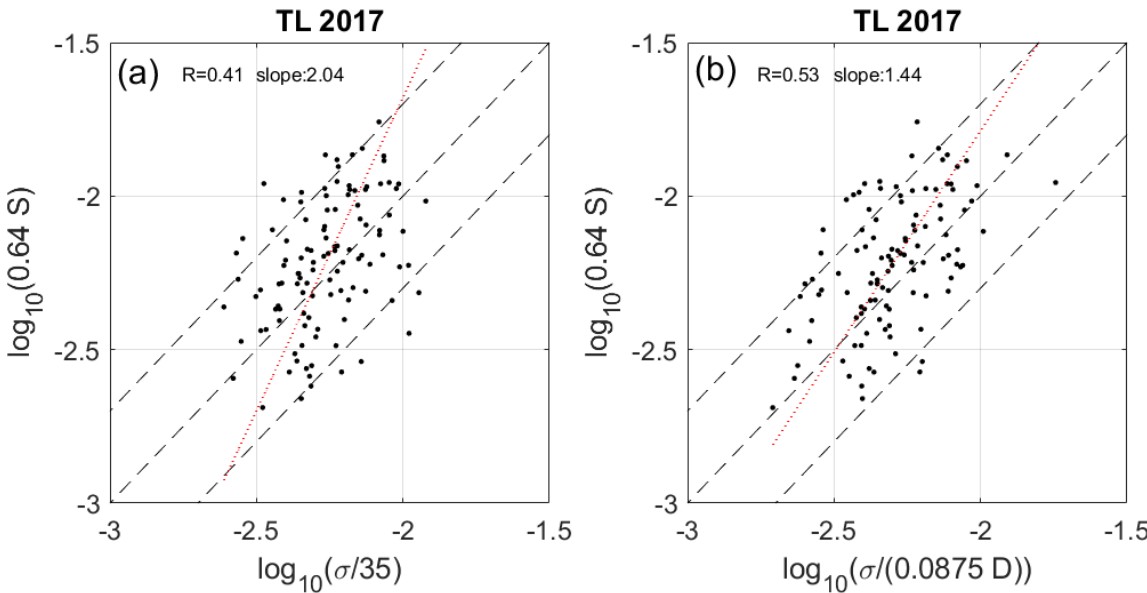

**Figure 11. (a) Same as in Fig. 6a for 113 arbitrarily selected layers in 2017 with $L_{out}$ = 35 m. (b) Same as (a) with $L_{out} = 0.0875\ D$.**

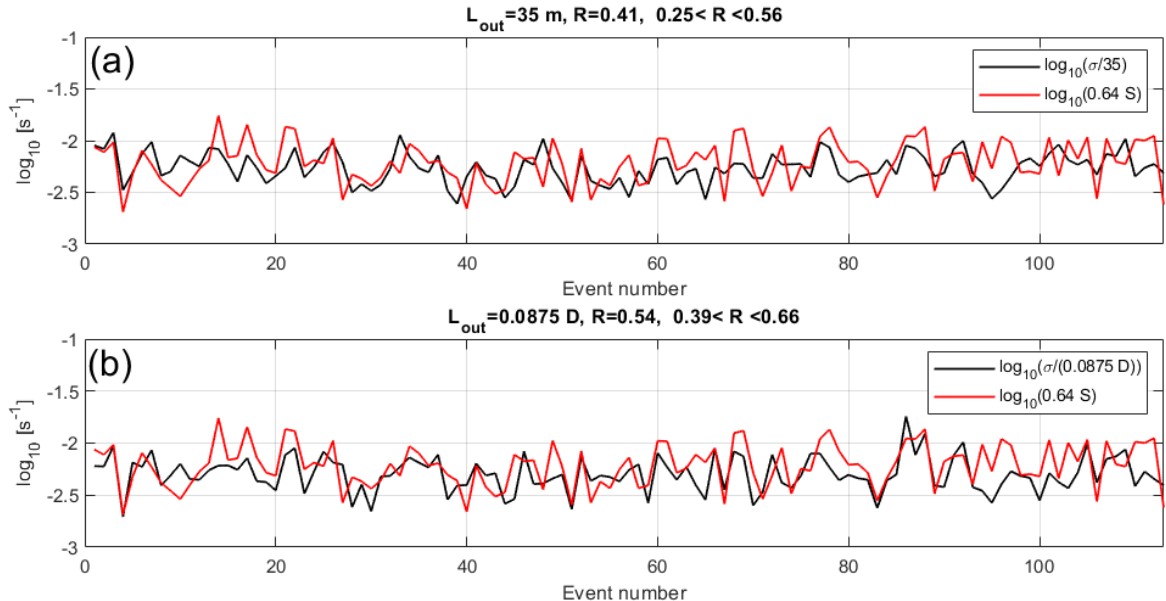

**Figure 12. Same as in Fig. 7, but for 113 arbitrarily selected layers.**





It was verified that comparisons between $\sigma$ and $S$ for layers associated with e.g., rain echoes, convective boundary layers and convection in clouds do not reveal (not shown) similar trends and significant correlations, indicating that the results described for KH layers correspond to physical properties of turbulence.

**6. Conclusions**

In the present work, we checked the relevance of a radar model $\varepsilon_S$ of TKE dissipation rate expected to be applicable for weakly stratified or strongly sheared layers. This model predicts a $\sigma^2 S$ dependence for low $Ri$ values, while the commonly used model $\varepsilon_N$ for stably stratified conditions predicts a $\sigma^2 N$ dependence. The latter was derived from multiple assumptions on the properties of the buoyancy subrange (e.g. *Hocking et al.*, 2016), but without explicit assumptions on $Ri$. The derivations by

*Basu and Holtslag* (2021) showed that both models are nearly quantitatively equivalent for $Ri \sim 1$ only, despite the fact that the two models are based on very different assumptions (see L2023 for more details). Because intense turbulence is expected to be observed for low $Ri$ values, the $\varepsilon_N$ model should underestimate the TKE dissipation rate when $Ri$ is large, a result consistent with the comparisons made with *in-situ* measurements by *Luce et al.* (2018) and L2023. Applied to turbulence generated by KH instabilities, the $\varepsilon_S$ model was found to be consistent with the $\varepsilon_{Lout}$ model predicting a $\sigma^3$ dependence, but

to a first approximation only, because the consistency was found to be more significant with a model predicting a $\sigma^3/D$ dependence, compatible with basic models of turbulence in nearly neutral boundary layers. This result suggests that similar dynamics occur in KH layers. As a corollary, the Hunt scale defined as $L_H = \sigma/S$ is a more appropriate scale than the buoyancy scale $L_B = \sigma/N$ to define the typical turbulence length scale in the observed KH layers, because $Ri$ values are small. We found $L_H \approx 0.056\, D$. The statistics made by L2023 showed that $\varepsilon_S$ cannot be used as the model by default as the condition of validity

of the model (low $Ri$ values) is not verified in the whole column of the atmosphere.

The main criterion for the validity of $\varepsilon_S$ is low $Ri$. Therefore, wherever this condition is met, $\varepsilon_S$ should be valid, even for stratospheric turbulence, where the background stratification is typically about four times more stable. Many observations suggest the existence of anisotropic turbulence in a stable stratified environment such as the stratosphere. It is also one of the most widespread hypotheses to explain the angular dependence of VHF radar echoes (*Hocking et al.*, 1986). This anisotropy

can only be explained by the influence of the stable stratification which inhibits the vertical component of turbulence. Therefore, the $\varepsilon_S$ model may not be valid in such circumstances. Further studies are needed to check the relevance of the models in the stratosphere with VHF radars.

**Appendix**: Derivation of the expression for heat diffusivity at small Ri values.


The eddy coefficient for heat or eddy diffusivity is given by (e.g. Lilly et al., 1974):

$$K_H = \gamma \frac{\varepsilon}{N^2} \qquad (A1)$$





where $\gamma = R_f/(1 - R_f)$ is the mixing coefficient defined as the ratio between the dissipation rates of potential and kinetic

energies. Using $\varepsilon_N = C_N \langle w'^2 \rangle N$ , we get the standard expression:

$$K_H = C \frac{\langle w'^2 \rangle}{N} \qquad (A2)$$

where $C = C_N \gamma$ is ~0.16 if $C_N \sim 0.5$ and $R_f = 0.25$ as often arbitrarily assumed in the literature (e.g. Lilly et al., 1974; Fukao

et al., 1994; Kurosaki et al., 1996; Naström and Eaton, 1997; Rao et al., 2001). Note that the arbitrary choice of $R_f$ is made to

avoid an inconsistency, when $R_f$ and $N$ go to 0, under neutral stratification. This inconsistency is removed when using

Expression (2) valid for small Ri values. Indeed, when introduced in (A1), we get:

$$K_H = \frac{0.64}{P_r \left(1 - R_f\right)^{1/2}} \frac{\langle w'^2 \rangle}{S} \qquad (A3)$$

which becomes

$$K_H = 0.8 \frac{\langle w'^2 \rangle}{S} \qquad (A4)$$

when $R_f$ goes to 0, since the turbulent Prandtl number $P_r$ goes to 0.8 for small Ri values. Note that 0.8 is a true constant as

long as the stationarity assumption remains true, unlike $C$ in (A2). Expression (A4) is thus an alternative expression for (A2)

for weakly stratified/strongly sheared conditions. Like $\varepsilon_S$ (Expression 2), Expression (A4) is independent of $N$ and can be

readily estimated from radar measurements of $\langle w'^2 \rangle$ and $S$. (A4) and (A2) are quantitatively identical when $Ri \approx 0.04$ and

(A2) leads to $K_H$ values ~2.5 times smaller than (A4) when $Ri \approx 0.25$.

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
