# Peer review of "Statistical assessment of a Doppler radar model of TKE dissipation rate for low Richardson numbers (weakly stratified or strongly sheared conditions)"

_Atmospheric Measurement Techniques, 2023_

## Referee Comment (RC1)

General comments:

This manuscript presents an impressive new report on assessment of a Doppler radar model of TKE dissipation rate for low Richardson numbers with a UHF wind profiler. The presented work provides a physical interpretation of $\varepsilon Lout$, which would be qualitatively identical to that for neutral boundary layers. The model is well interpreted and studied. Thus, the manuscript is recommended to be published with minor revision.

Specific comments:

1. The quality of Figure 2 should be improved, such as font size and color bar.
2. Compared with stable conditions, weakly stratified or strongly sheared conditions have different turbulent scale. A discussion about relevance to the resolution of the detection instrument is recommended to added, for resolution of wind profiler is 1 min/100 m in this paper.
3. In section of Conclusion, the authors might need to address in more details some limitations in the present study including data and methodology.

---

## Author Comment (AC1)

**Reviewer 1**

**General comments :**

This manuscript presents an impressive new report on assessment of a Doppler radar model of TKE dissipation rate for low Richardson numbers with a UHF wind profiler. The presented work provides a physical interpretation of $\varepsilon Lout$, which would be qualitatively identical to that for neutral boundary layers. The model is well interpreted and studied. Thus, the manuscript is recommended to be published with minor revision.

We thank the reviewer for his comments.

Specific comments :

1. The quality of Figure 2 should be improved, such as font size and color bar.

Figure 2 has been redrawn for a better clarity and uniform notations.

2. Compared with stable conditions, weakly stratified or strongly sheared conditions have different turbulent scale. A discussion about relevance to the resolution of the detection instrument is recommended to added, for resolution of wind profiler is 1 min/100 m in this paper.

A discussion related to the radar limitations has been added in a new section 6 (Discussion), also required by reviewer 2.

3. In section of Conclusion, the authors might need to address in more details some limitations in the present study including data and methodology

We also included a few sentences on this topic, which continues the point made in 2).

---

## Author Comment (AC2)

**Reviewer 2**

This paper presents a feasible method to estimate TKE dissipation rate (ε) based on a unit of WPR-LQ-7 in a sheared atmospheric boundary layer. The model is illustrated clearly with proper experimental settings. The authors point out the relevance of $L_{out}$ and $D$, which is valuable in a real-world application. Although the condition (in a sheared environment) is a little bit strict, I think this work can be published after some minor revisions.

We thank the reviewer for his comments. Please note that the analysis was made from data collected above the atmospheric boundary layer. It does not include KH instabilities that may develop at the CBL top, for example, to avoid cases that may interact with other instabilities.

Specific comments:

1. In **Abstract**, the definition of $L_{out}$ should be clearly pointed out.

   Done. $L_{out}$ refers to an apparent outer scale estimated from radar Doppler variances and UAV-derived TKE dissipation rates (Luce et al., 2018). This scale was noted $L_c$ by Luce et al. (2018) ('c' stands for 'characteristic'). But the term $L_{out}$ is perhaps more adapted because it can be related to turbulence scales as shown in the present work.

2. Line 30: *half the Doppler spectral width* mentioned here is not consistent with *half width σ of the Doppler spectrum peak* mentioned in **Abstract**, which may confuse readers without background.

   We used the terminology "*half the Doppler spectral width*" *everywhere*. The spectral width is 2σ (twice the standard deviation)

3. Line 42-46. The sentence starting from *(2) The model...* is too long. I suggest the authors use short sentences.

   We have modified the text as follows:

   The model $\varepsilon_S = C_S \sigma^2 S$ provides better agreement than $\varepsilon_N = C_N \sigma^2 N$. $\varepsilon_N$ is commonly used by the MST radar community. It is expected to be applicable to turbulence under stable stratification *(e.g. Hocking, 1983; Hocking et al., 2016)*. $\varepsilon_S$ was originally proposed by *Hunt et al. (1988)* from heuristic arguments and confirmed by *Basu et al. (2021)* and *Basu and Holtslag (2022)* from Direct Numerical Simulations (DNS) and analytical derivations. It is expected to be valid for weakly stratified and/or strongly sheared conditions, i.e. for low *Ri* values.

   We have also revised the text to avoid sentences that are too long in some places.

4. In Table 1, the authors list many key parameters of the WPR-LQ-7. It would be better if they could include what parameters can be derived from WPR-LQ-7 measurement. In this section, I cannot find this critical information.

   It is included in the text and in the table 1.

5. Line 123. The authors need to explain why a KH layer can be visible with enhanced SNR signal or to provide a reference.

It is now explained how a KH layer can be identified from its distinctive signature in time-height cross-sections of radar echo power

6.  In Figure 3, I suggest the authors mark the KH layer mentioned in Figure 2a with two dash lines or a grey zone.

    Done.

7.  Line 304-307. This part should be mentioned in a **Discussion. I suggest** the authors discuss the strengths and shortages of the new mode compared with other studies in this paragraph and change the section title to **Discussion**.

    We included a section "Discussion" before the conclusion.